# Comprehensive Fertility Management After Pituitary Adenoma Surgery: Lessons from a Rural Japanese Case and Practical Review

**DOI:** 10.3390/reports8030144

**Published:** 2025-08-15

**Authors:** Daisuke Numahata, Kosuke Kojo, San-e Ishikawa, Takumi Kuramae, Ayumi Nakazono, Kaoru Yanagida, Hiroyuki Nishiyama, Tatsuya Takayama

**Affiliations:** 1Center for Human Reproduction, International University of Health and Welfare Hospital, 537-3, Iguchi, Nasushiobara, Tochigi 329-2763, Japantakayama@iuhw.ac.jp (T.T.); 2Department of Urology, International University of Health and Welfare Hospital, 537-3, Iguchi, Nasushiobara, Tochigi 329-2763, Japan; 3Department of Urology, Institute of Medicine, University of Tsukuba, 1-1-1, Tennodai, Tsukuba, Ibaraki 305-8575, Japan; 4Division of Endocrinology and Metabolism, Department of Medicine, Jichi Medical University, 3311-1, Shimotsuke, Tochigi 329-0498, Japan; saneiskw@jichi.ac.jp; 5Department of Endocrinology and Metabolism, International University of Health and Welfare Hospital, 537-3, Iguchi, Nasushiobara, Tochigi 329-2763, Japan; 6Department of Neurosurgery, National Hospital Organisation Tochigi Medical Center, 1-10-37, Nakatomatsuri, Utsunomiya, Tochigi 320-8580, Japan

**Keywords:** benign brain tumor, central nervous system, CNS tumor, fertility preservation, male infertility counseling, oncofertility care network, PitNET, reproductive endocrinology, sexual complications of neurosurgery, sperm cryopreservation

## Abstract

**Background and Clinical Significance:** Pituitary adenomas, also termed pituitary neuroendocrine tumors, pose a significant risk of hypogonadotropic hypogonadism (HH) after surgical resection, with profound consequences for fertility and sexual function in young patients. **Case Presentation:** We present the case of a 29-year-old man from rural Japan who developed severe HH and azoospermia following two transsphenoidal resections for a large pituitary adenoma. Despite early engagement with neurosurgery teams, fertility management was delayed by the absence of on-site endocrinology expertise and limited local oncofertility resources. After comprehensive endocrine evaluation and counseling, the patient began combined human chorionic gonadotropin and recombinant follicle-stimulating hormone therapy, resulting in full recovery of sexual function and normalization of semen parameters, ultimately leading to spontaneous conception and the birth of a healthy child. Building on this real-world case, we provide a narrative review of current practical management strategies for HH after pituitary surgery, including the utility of hormone-stimulation tests, Japanese guideline-based subsidy systems, and best-practice approaches to hormonal replacement. **Conclusions:** This case underscores not only the necessity for early, interdisciplinary collaboration and preoperative counseling but also highlights a rare instance in which a patient with a benign tumor received care that did not address his fertility-related needs, emphasizing that such considerations should be integrated into preoperative counseling even for non-malignant conditions. Strengthening regional oncofertility networks and improving healthcare providers’ awareness of fertility-preservation options remain essential for improving outcomes.

## 1. Introduction and Clinical Significance

Tumors of the pituitary and sellar region range from highly malignant germ cell tumors to benign lesions such as craniopharyngiomas, with pituitary adenomas—benign neuroendocrine neoplasms confined to the sella—being the most representative [1]. Pituitary tumors, whether benign or malignant, pose a significant risk of gonadotropin deficiency following surgical intervention [2,3]. The resulting hypogonadotropic hypogonadism (HH) can lead to decreased libido, erectile and ejaculatory dysfunction, and impaired spermatogenesis, potentially resulting in infertility. For young patients who desire children, these consequences can be emotionally and socially distressing.

Despite advances in fertility preservation, many patients have historically not received adequate counseling regarding the risk of permanent or transient infertility before undergoing tumor resection [4]. This gap in clinical care is particularly evident in rural settings, where a lack of specialist networks often limits access to comprehensive oncofertility services [5].

In recent years, multidisciplinary collaboration among oncologists, neurosurgeons, and reproductive specialists has been increasingly recognized as essential [6,7,8]. However, awareness of fertility-preservation guidelines among neurosurgeons and other non-reproductive specialists remains inconsistent, leading to missed opportunities for preoperative counseling and fertility preservation. While HH after pituitary surgery is common and may be considered a routine complication in daily practice, this case illustrates how, even in such ordinary cases, best-practice oncofertiliity care is not always delivered. In this case report, we present a patient in Japan who underwent surgical resection of a pituitary adenoma without receiving any fertility-related counseling. Postoperatively, he developed HH and later sought assistance at a specialized reproductive center. Building on this real-world case, we provide a detailed narrative review of the clinical management, diagnostic approach, and the relevant literature, aiming to illustrate both the limitations encountered in routine practice and the potential for best-practice care. Through this case, we aim to emphasize the importance of comprehensive preoperative infertility discussions and highlight the challenges posed by limited oncofertility resources in certain regions. In particular, we present a seemingly rare example described in the literature in which a patient with a benign pituitary lesion experienced care that failed to address both his clinical needs and personal priorities regarding fertility, thereby underscoring that even in non-malignant conditions, pre-treatment fertility considerations warrant the same level of attention as in oncology settings.

## 2. Case Presentation

The patient was a 29-year-old man with no significant medical or family history. He initially presented to a local ophthalmologist with visual disturbances as the chief complaint, raising suspicion of a pituitary tumor, and was referred to a local neurosurgeon for primary care. He was then referred to the neurosurgery department of a core regional hospital, where brain magnetic resonance imaging (MRI) revealed a 27 × 34 mm pituitary tumor, primarily located within the sella turcica, compressing the optic nerve superiorly, but with no evidence of lateral invasion into the cavernous sinus. The tumor showed isointensity on T1-weighted images, high intensity on T2-weighted images, and marked contrast enhancement, which were typical findings of a pituitary adenoma [9] (Figure 1a). Preoperative hormonal assessments showed that prolactin (PRL) and follicle-stimulating hormone (FSH) were above the upper limit of their reference intervals; however, no overt related clinical symptoms existed, and the tumor behaved as a “clinically non-functioning” adenoma [10]. In fact, no significant abnormalities were observed in any other endocrine axes except for PRL and FSH; however, testosterone and other gonadal hormones were not evaluated preoperatively. Based on these findings, a preoperative diagnosis of a clinically non-functioning pituitary adenoma was made. Notably, testosterone was not evaluated preoperatively (Table 1). It should be noted that “pituitary adenoma” is the conventional name for what is now also termed a pituitary neuroendocrine tumor (PitNET), and these terms are used interchangeably [11]. In this report, “pituitary adenoma” is used throughout unless otherwise specified.

A detailed timeline of the patient’s clinical course—including the stages of pituitary tumor management at a regional hospital, manifestation of sexual dysfunction, and the referral process, as well as specialized reproductive endocrine care at our institution—is presented in Figure 2. The patient’s tumor was compressing the optic nerve, and urgent surgery was planned without delay. The patient underwent initial tumor resection via a transnasal-sphenoidal approach. The tumor was soft and could be easily removed by suction alone. During surgery, the normal pituitary gland was also visualized and therefore preserved. Postoperative computed tomography (CT) and MRI showed that approximately 80% of the tumor had been resected and decompression of the optic nerve was achieved; however, a residual tumor remained on the left side (Figure 1b). Therefore, a second surgery was performed 14 days after the initial surgery, using the same approach, but with a large fenestration of the left sellar floor to provide an optimal, tumor-centered view and facilitate complete removal of the remaining tumor, with the outer dura and superior arachnoid clearly visualized. On CT performed 1 day after the second surgery, although postoperative changes were observed, such as hemorrhage and edema, no obvious residual tumor was seen. Subsequent MRI performed after sufficient time had passed since the second surgery demonstrated complete tumor removal (Figure 1c). Stereotactic radiosurgery, including Gamma Knife radiosurgery, is considered a primary treatment option when surgical resection is contraindicated [12]. In particular, for residual nonfunctioning pituitary adenomas, it has been suggested as a valuable alternative to avoid repeat surgery [13]. Nevertheless, in this case, surgical resection was selected as the first-line treatment in accordance with standard principles widely adopted in Japan [14].

Histopathological examination of the resected specimen with hematoxylin and eosin staining revealed a proliferation of nearly uniform, round-to-polygonal, predominantly chromophobic cells, leading to a diagnosis of pituitary adenoma according to the conventional classification. Immunohistochemistry of the tumor was positive for FSH and luteinizing hormone (LH), and negative for growth hormone (GH), PRL, thyroid-stimulating hormone (TSH), and adrenocorticotropic hormone (ACTH), corresponding to gonadotroph adenoma in the previous classification, or SF1-lineage PitNET—a tumor entity characterized by expression of steroidogenic factor 1 (SF1)—according to the fifth edition of the “World Health Organization (WHO) Classification of Endocrine and Neuroendocrine Tumors” (2022) [15] and the fifth edition of the *General Rules for Clinical and Pathological Studies on Brain Tumors* (2023) by the Japan Neurosurgical Society [16].

Due to concerns about central diabetes insipidus resulting from arginine vasopressin deficiency, permanent oral desmopressin replacement was started on postoperative day 2 after the first surgery, without further evaluation. Although no overt signs of adrenal insufficiency appeared, both ACTH and cortisol levels fell below the lower limit of their reference intervals postoperatively. Fortunately, except for the serious problems with sexual function described later, no signs of life-threatening endocrine complications during the perioperative period were observed. At that time, other potential endocrine disorders were overlooked and not thoroughly evaluated (Table 1).

Reflecting on his preoperative memory, the patient reported that he had no concerns regarding erection, ejaculation, sexual intercourse, or masturbation prior to surgery. He had already been engaged to his 27-year-old partner at that time, but his illness had delayed their marriage plans. Less than 1 month after the successful second surgery, they were able to complete their marriage registration and preparations for living together. Although the couple had not set a concrete schedule for having children, they strongly wished to have children in the future. However, during the same period, he began to notice decreased libido and erectile dysfunction after surgery, making sexual intercourse impossible. In an attempt to confirm his fertility, he engaged in prolonged masturbation but found that he produced little or no semen. This experience was not only distressing in itself but also led to deep anxiety regarding his future fertility. These symptoms became apparent soon after resuming married life, but he did not immediately share his concerns with his neurosurgeon. Ultimately, it was not until a follow-up visit 89 days after the second surgery that the neurosurgeon fully recognized the patient’s concerns. At this point, his testosterone was measured for the first time, revealing a level below the assay sensitivity and clearly indicating endocrine-related sexual dysfunction (Table 1). Around this time, because of concern about low cortisol levels, oral hydrocortisone was started 96 days after the second surgery. However, as expected, this alone did not improve his sexual dysfunction, and the neurosurgeon engaged in thorough discussions with an in-house urologist. Nevertheless, despite these efforts, an optimal endocrinological approach was not implemented, partly due to the absence of a full-time endocrinologist at the regional hospital. Recognizing these challenges, a hospital staff member sent an email to the publicly available address of our infertility consultation service, as listed on our institution’s website, 117 days after the second surgery. This direct outreach promptly initiated a review of the case and consideration of treatment strategies by our urology team specializing in reproductive endocrinology.

Specialized endocrine evaluations, including stimulation tests, were considered essential not only for accurately assessing the patient’s endocrinological condition and developing an effective treatment strategy [17,18], but also because such evaluations are mandatory for applying for subsidies for the expensive medications likely to be introduced [19]. Table 2 shows representative guidelines related to HH published in Japan. Among these, meeting the diagnostic criteria based on the guidelines developed by the Survey and Research Group for Hypothalamic-Pituitary Dysfunction (Health and Labour Sciences Research Grant for Research on Rare and Intractable Diseases) is a prerequisite for receiving subsidies.

Therefore, our team believed that the patient should be referred to our institution as soon as possible. However, as is often the case in many Japanese medical centers, when we received the initial email inquiry, our hospital was about to enter an extended year-end and New Year holiday closure. In Japan, the 6-day period from December 29 to January 3 is legally designated as a non-working period for governmental institutions [31], and most private companies follow the same schedule [32], with many hospitals being no exception [33]. When Saturdays and Sundays are adjacent to this period, the closure is further extended [34,35]. Although maintaining medical resources during holidays equivalent to those on weekdays is considered important [36], it is well known that delays in diagnosis and initiation of treatment frequently occur during such holiday closures [37], and in fact, increased clinical risks during the New Year period in Japan have been reported [38]. Moreover, the patient was expected to become extremely busy with work after the holidays, making it difficult for him to take time off for evaluation. Given that our institution was located some distance from the patient’s place of residence and that outpatient visits would be burdensome, the attending neurosurgeon at the regional hospital, determined to address the patient’s concerns proactively, proposed that the patient continue outpatient visits to the regional hospital during its year-end closure and personally made the effort to conduct the necessary evaluations in-house. The patient agreed to this plan.

We maintained close collaboration with the regional hospital team via fax, sending them a manual created by our team in accordance with standard protocols, and had them conduct the endocrine stimulation tests as promptly as possible at their hospital. Although the patient’s main concerns were limited to the hypothalamic–pituitary–gonadal axis, we considered that dysfunction could involve the entire hypothalamic–pituitary axis. Several protocols for endocrine management after pituitary surgery have been proposed [39,40]. In this case, stimulation tests for GH, PRL, TSH, ACTH, and gonadotropins were performed 133 days after the second surgery, according to standard protocols widely used in Japan [41,42,43].

Table 3 shows the time course of anterior pituitary hormone concentrations in response to various stimulation tests. These results are consistently explained by pituitary damage after tumor resection and/or sequelae of hypothalamic compression by the tumor, without contradiction—even for the paradoxically mildly elevated PRL, as well as the persistently and markedly low baseline levels of GH, TSH, LH, and FSH, and the mildly reduced ACTH observed after two pituitary tumor resections. At this point, oral desmopressin replacement had already been initiated, and since no problems related to central diabetes insipidus were observed, an endocrinological evaluation of the posterior pituitary function was omitted.

Although secondary hypogonadism due to pituitary tumor resection was confirmed in this case, the possibility of concomitant primary hypogonadism could not be excluded. Therefore, a standard human chorionic gonadotropin (hCG) stimulation test, widely used in Japan, was performed [55,56,57,58,59,60]. As shown in Table 4, testicular testosterone secretion in response to hCG administration was favorable, suggesting that hCG would be effective as part of the treatment strategy. Notably, the hCG stimulation test was a mandatory requirement for obtaining subsidies according to the 2010 guidelines, and is mentioned as a reference finding in the 2019 and 2023 guidelines shown in Table 2 [20,21,22]. In addition to testosterone produced by Leydig cells in the testicular interstitium, anti-Müllerian hormone (AMH) and inhibin B—both secreted by Sertoli cells that support the seminiferous tubules and whose secretion is stimulated by FSH—are representative markers that decrease in response to hypogonadism and are useful for assessing its pathophysiology [61]. However, AMH cannot be measured under the Japanese health-insurance system, and inhibin B is not available as a routine laboratory test in Japan [27,30]. As a result, neither hormone is considered a standard parameter in clinical practice, and thus, they were not measured in this case.

As the year-end and New Year holidays were approaching, we arranged for the patient to visit our institution for a one-time, in-depth interview and counseling session immediately after the scheduled stimulation tests were completed—that is, 136 days after the second surgery. At this session, although not all test results were available, we were able to provide counseling regarding the significance of the several days of endocrine testing performed at the previous hospital, the anticipated results and corresponding fertility-treatment strategies, and guidance on subsequent management procedures—which would have been difficult for an inexperienced neurosurgeon to manage alone.

On the same day as the initial counseling session, physical examination revealed a height of 175 cm, a weight of 71 kg, and a body mass index of 23.2 kg/m^2^. Assessment of pubic-hair distribution by visual inspection was consistent with Tanner stage 5 (P5), and external genitalia were also consistent with Tanner stage 5 (G5) [65]. However, testicular volumes measured with a punched-out orchidometer were 16 mL bilaterally [66], corresponding to Tanner stage 4 (G4). No palpable varicocele was detected in the standing position, even during a concurrent Valsalva maneuver [67]. Ultrasonography showed no testicular tumors, and pelvic MRI revealed no abnormalities (Figure 3).

On the same day as the initial counseling session, baseline sexual function after two pituitary tumor resections was assessed using questionnaires (Table 5). Notably, symptoms associated with testosterone deficiency as evaluated by the Heinemann Aging Male’s Symptoms (AMS) scale were classified as severe only in the sexual subscale, while the somatic and psychological subscales were assessed as none or mild. The presence of both decreased libido and erectile dysfunction at this point was quantified by scores of 4 (severe) for both item 17 (disturbed libido) and item 18 (impaired sexual potency) of the AMS, and a score of 1 on the Erection Hardness Score (EHS), which indicates erection hardness at sexual stimulation as larger but not hard. To objectively assess erectile function, we attempted to evaluate nocturnal penile tumescence using a dedicated device consisting of a sliding band and plastic cursor (Erectometer, Kysmaq, Tokyo, Japan) [68]. However, the device could not be kept in place as it came off easily, and therefore, the assessment could not be completed. Additionally, the patient responded with a score of 1 (“could not ejaculate”) to the question regarding frequency of ejaculation, indicating self-awareness of ejaculatory dysfunction. According to Otani’s classification, which is widely used in Japan, this corresponds to “I-(A): both ejaculation by masturbation and intravaginal ejaculation are impossible” [69].

Based on the detailed medical interview, physical examination, endocrinological evaluation, and imaging studies described above, the patient’s acquired sexual dysfunction was diagnosed as being caused by panhypopituitarism. At the time of the initial counseling, oral desmopressin and hydrocortisone replacement had already been started by the referring physician; however, since concomitant adrenal and thyroid insufficiency were subsequently confirmed, levothyroxine was added, and the doses of both hydrocortisone and levothyroxine were continuously adjusted. The severe sexual dysfunction that developed after two pituitary resections caused significant anxiety for the patient, but most of this was alleviated during the initial counseling session. As a result, both the patient and his wife expressed a strong desire to pursue conception using methods as natural and low-cost as possible, even if it took additional time. Therefore, for hormone replacement therapy (HRT) for LH and FSH deficiency, as well as for GH deficiency, it was agreed that the former would be initiated only after approval of the governmental subsidy, and the latter would be postponed and introduced promptly only if typical symptoms such as easy fatigability or impaired concentration became apparent [78], in order to minimize financial burden. Since the patient met the diagnostic criteria for hypopituitarism as designated intractable disease No. 78 by the Japanese Ministry of Health, Labour and Welfare, the application for medical expense subsidies under the designated intractable disease program was completed, and HRT was initiated 192 days after the second pituitary surgery (56 days after the initial counseling at our institution; Figure 2).

The content of HRT and the trends in laboratory findings are presented in Figure 2. For LH and FSH deficiency, combination therapy with self-injected hCG and recombinant follicle-stimulating hormone (rFSH) is considered the most effective and ideal treatment [79]. However, since rFSH is considerably more expensive than hCG, a commonly practiced approach in Japan is to initiate treatment with hCG monotherapy and add rFSH only if spermatogenesis does not sufficiently recover within 6 months [55]. To assess baseline spermatogenesis, a semen analysis was attempted on the day HRT was initiated. However, despite prolonged attempts at masturbation, the patient was unable to achieve ejaculation, and microscopic examination of the small amount of secretion obtained revealed no spermatozoa. This was clearly due to ejaculatory dysfunction, and retrograde ejaculation could not be ruled out. Nevertheless, given the long-standing FSH deficiency, even if retrograde ejaculation had been present, spermatogenesis was presumed inactive; therefore, an invasive search for spermatozoa in the bladder was not performed [80].

At 14 days after initiation of HRT, TT had increased to 5.98 ng/mL, and the T/E ratio was 18.9 (ng/dL)/(pg/mL), both showing marked improvement. At this point, the T/E ratio exceeded the target value of 12.0 (ng/dL)/(pg/mL) [64]. Erectile function, as assessed by the EHS, may have also improved slightly to EHS 2 at the same time; however, the patient reported continued difficulty with ejaculation, and therefore, semen analysis was canceled. At 42 days after starting HRT, erectile function had fully recovered to EHS 4. Although azoospermia persisted on semen analysis, the semen volume was 4.3 mL, indicating restoration of ejaculatory function. At 84 days after starting HRT, a very small number of spermatozoa were detected for the first time on semen analysis. The patient was informed of the option to switch from hCG replacement—which requires multiple self-injections per week—to testosterone replacement therapy with less frequent injections, using cryopreserved ejaculated sperm or testicular sperm extraction (TESE). However, the patient elected to continue the current treatment regimen in order to pursue natural conception. At this point, AMS had improved to 17, indicating complete resolution of sexual dysfunction (Table 5).

At 140 days after initiating HRT, since sperm concentration had not sufficiently recovered, rFSH was added to the regimen. At 210 days, semen analysis results exceeded the reference values for the first time, with a semen volume of 3.8 mL, a sperm concentration of 73 × 10^6^/mL, and sperm motility of 73.0% (reference values: 1.4 mL, 16 × 10^6^/mL, and 42%, respectively) [81]. At 329 days, the patient’s wife was confirmed to be 9 weeks pregnant following spontaneous conception. The pregnancy progressed uneventfully, and at 545 days after initiating HRT, a healthy baby without congenital anomalies was delivered at 40 weeks and 6 days of gestation (Figure 2).

As the patient expressed a desire to conceive a second child naturally in the near future, he opted to continue HRT and regular semen analyses during his wife’s pregnancy, rather than cryopreserve ejaculated sperm. To reduce the burden of frequent self-injections, the intensity of HRT was gradually tapered, with regular semen-analysis monitoring. Adjustments to HRT intensity were also aimed at reducing E2 levels, which often exceeded the upper limit of the reference interval (48.8 pg/mL) due to hCG stimulation, given concerns that E2 excess may diminish therapeutic efficacy [82,83] and increase the risk of thrombosis [84], prostate cancer [85], and cardiovascular disease [86]. After pregnancy was confirmed, the HRT intensity was gradually reduced, but no trend toward deterioration in either sexual function or semen parameters was observed (Figure 2). Throughout the observation period following HRT initiation, no adverse events were observed, including gynecomastia, erythrocytosis, acne, sleep apnea associated with increased testosterone levels [87], or injection-site pain attributable to hCG administration [88].

The patient reviewed the content of the original draft and provided written informed consent for publication.

## 3. Discussion

A key issue highlighted by this case—postoperative spermatogenic dysfunction and sexual dysfunction following pituitary adenoma resection—is the lack of adequate preoperative counseling regarding the risk of gonadotropin deficiency. Several factors contribute to this gap in care. First, neurosurgeons who do not specialize in pituitary disorders often have limited knowledge of reproductive endocrinology and may be unaware of fertility-preservation considerations that are standard in oncologic care. Second, while interdisciplinary collaboration between reproductive specialists and oncologists has gained recognition, some healthcare regions have underdeveloped referral networks, creating further barriers to timely fertility counseling and preservation.

In 2006, the American Society of Clinical Oncology (ASCO) published guidelines recommending that physicians inform reproductive-age patients and parents of pediatric patients about the risk of fertility loss and available fertility preservation options in the context of cancer treatment [89]. These recommendations were reaffirmed in 2013 [90] and 2018 [91]. In Japan, the Japan Society of Clinical Oncology (JSCO) released its own guidelines in 2017, addressing fertility preservation in childhood, adolescent, and young-adult patients with cancer [92]. However, physician awareness of fertility preservation remains inconsistent, and some reproductive-age patients begin oncologic treatment without any discussion of fertility-related risks or options [93].

Brain or other central nervous system (CNS) tumors encompass a wide spectrum of pathologies [94], each of which may require different combinations of surgery, radiotherapy, and chemotherapy [95]. In cases involving elevated intracranial pressure or neurological deficits, there may be little time or insufficient clinical stability to address fertility concerns before treatment. Additionally, a poor long-term prognosis may deter physicians from introducing fertility-preservation discussions. However, a 2024 systematic review demonstrated that patients with neuro-oncological diseases can successfully undergo fertility preservation and achieve healthy live births using stored specimens [96]. A comprehensive 2019 study in Japan found that 2.6% of patients who successfully underwent sperm cryopreservation had brain tumors [97].

While much of the literature focuses on chemotherapy-induced fertility risks in malignant tumors, it is equally important to recognize the essential role of the pituitary gland in reproductive function. Highly invasive malignant tumors or inflammatory lesions of the pituitary can directly cause hormonal deficiencies, including hypogonadism. However, symptoms such as fatigue are often nonspecific, allowing hypogonadism and subsequent azoospermia to go undiagnosed without formal testing [98]. Benign lesions, such as pituitary adenomas, typically progress slowly, and smaller lesions may not cause hypopituitarism. This can lead to an underestimation of fertility risks. Nonetheless, surgery or radiotherapy for pituitary adenomas can result in hypopituitarism, even though the adenoma itself rarely causes pituitary failure preoperatively.

Not all patients undergoing pituitary tumor surgery develop gonadotropin deficiency; the risk depends on tumor type and the invasiveness of treatment. The incidence of hypogonadism after resection of non-functioning pituitary adenomas has been reported at approximately 8.3% [99]. Although this figure is not exceedingly high, it remains clinically significant. Additionally, patients with pituitary disorders may already have impaired semen quality and low androgen levels before treatment [100].

Compared to patients receiving chemotherapy for malignancies, the perceived urgency of fertility preservation in pituitary surgery may be lower, as pituitary-targeted surgery or radiotherapy does not directly damage testicular tissue. Primary testicular injury from chemotherapy is often permanent [101], whereas secondary testicular dysfunction due to pituitary insufficiency may be reversible with gonadotropin replacement therapy [102]. Nonetheless, preoperative disclosure of the risk of HH remains essential. Even if infertility is temporary, couples may experience significant distress from an unanticipated interruption in spermatogenesis, especially given the limited window of female reproductive potential. Moreover, fertility preservation offers psychological benefits, including a preserved sense of identity and reduced regret [103]. For patients without confirmed normal semen parameters or prior successful fatherhood, gonadotropin therapy may take months to years and does not guarantee a successful return of sperm production [104]. This issue is particularly relevant in Japan, where delayed childbearing is increasingly common, making early intervention and potential sperm cryopreservation especially valuable. Case reports support the utility of sperm cryopreservation even in non-malignant conditions, such as inguinal hernia repair [105] and transurethral surgeries [106,107].

Preoperative sperm cryopreservation also broadens future therapeutic options. In addition to causing azoospermia, HH results in low testosterone, which can lead to decreased libido, erectile dysfunction, and ejaculatory disorders, all of which significantly impact quality of life [108]. While combined hCG and rFSH therapy effectively restores spermatogenesis and improves sexual function, it is prohibitively expensive for lifelong use [109]. In Japan, designated intractable diseases (nanbyo) qualify for governmental financial support [110], and HH is among these conditions [19]. However, if gonadotropin therapy becomes unsustainable due to financial strain or other disruptions (e.g., supply chain issues or limited access in certain countries), stored sperm would allow clinicians to focus on treating low testosterone symptoms alone. Testosterone replacement is a comparatively cheaper and more accessible therapy that effectively improves libido and erectile function, though it does not restore fertility [111]. Notably, the hCG used for HRT in this case is a drug manufactured using the classical principle of extraction from human urine [112]. Amidst the current global shortage of pharmaceuticals [113], the unstable procurement of hCG in Japan has been pointed out [55]. If hCG becomes unavailable, even in this case, after sperm cryopreservation, it may be necessary to switch from hCG-based HRT to testosterone replacement.

When managing pituitary adenoma cases in young male patients concerned about fertility, preoperative evaluation by a reproductive endocrinologist and consideration of fertility preservation remain prudent, even for benign lesions not requiring chemotherapy. In Japan, facilities offering sperm cryopreservation are listed on the Japan Society for Fertility Preservation website (https://www.j-sfp.org/network/ (accessed on 25 July 2025)), and those providing gonadotropin replacement therapy for male hypogonadism can be found via the Japan Society for Reproductive Medicine (http://www.jsrm.or.jp/document/danseifunin_enquete.pdf (accessed on 25 July 2025)). These resources help mitigate geographic barriers, although gaps persist. With increasing recognition of oncofertility consortia [114], regional oncofertility networks have been expanding throughout Japan [115], and healthcare professionals are gradually becoming more proactive in offering fertility preservation counseling before cancer treatments.

Another challenge is that discussing sexual function can be uncomfortable, and patients’ dysfunction may go undetected without direct questioning [116]. Physicians often feel uneasy initiating discussions about sexual health, lack confidence in managing sexual dysfunction, and face time constraints that limit in-depth assessments [117]. A survey of neurosurgeons specializing in spinal-cord disorders found that only 5% had ever referred a patient to a reproductive specialist, despite routinely encountering sexuality-related concerns [118]. Notably, more than half of the respondents expressed interest in improving their knowledge to better engage patients in discussions about fertility. Future efforts should focus on strengthening professional networks and developing educational programs for neurosurgeons, nurses, and allied health professionals to facilitate timely referrals [119].

At our institution, the infertility consultation service played a critical role in bridging the gap between the referring hospital and our fertility specialists, minimizing treatment delays. However, such cases remain the exception, raising concerns about unaddressed cases with similar risks. Fortunately, Japan—alongside Austria and Germany—maintains a well-established national oncofertility registry [120]. Maximizing the utility of such resources is essential to prevent delays in fertility preservation and ensure that patients who wish to have biological children do not miss this opportunity.

An important strength of this case report is that, although low awareness of fertility-preservation guidelines among non-reproductive specialists has been documented previously, to our knowledge there have been no prior case reports that describe an actual instance in which a patient experienced care that failed to address the patient’s clinical needs and personal priorities. Furthermore, unlike malignant tumors requiring consideration of chemotherapy-induced infertility, this case draws attention to pre-treatment fertility in benign pituitary lesions, an area that has been insufficiently documented to date. This emphasis underscores that, even for benign conditions, healthcare providers should recognize the necessity of appropriate preoperative counseling and remain mindful that patients may wish to pursue fertility preservation measures such as sperm cryopreservation. Nevertheless, this report has several limitations. First, although in this case the patient ultimately achieved fatherhood and, according to an interview, appeared satisfied with the final outcome, it describes the clinical course of a single patient and, by its nature, cannot clarify, for example, differences in patient satisfaction between those who did and did not receive preoperative counseling from the outset. Since deliberately withholding counseling would be ethically unacceptable in a prospective study, the accumulation of similar case reports is important. Second, certain aspects of this case are specific to the circumstances of rural Japan, and caution is warranted when generalizing the findings to other settings.

## 4. Conclusions

This case highlights how limited awareness of reproductive endocrinology among neurosurgeons, coupled with underdeveloped oncofertility networks in some regions, can leave young patients unprepared for the potential risk of gonadotropin deficiency after pituitary tumor resection. Nonetheless, early collaboration with specialized teams facilitated successful fertility management for this patient. Strengthening local oncofertility networks and expanding educational initiatives—particularly for healthcare providers who do not routinely manage reproductive-age patients with cancer—will be crucial to ensuring timely, comprehensive care in future cases.

## Figures and Tables

**Figure 1 reports-08-00144-f001:**
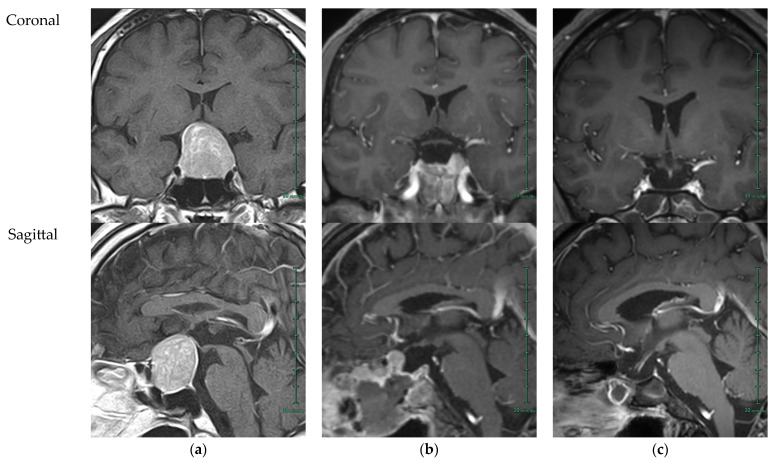
Sequential coronal and sagittal T1-weighted post-gadolinium enhanced magnetic resonance imaging (MRI) images of the hypothalamo-pituitary region before and after two transsphenoidal pituitary tumor resections. (**a**) MRI obtained 19 days before the initial pituitary tumor resection. A pituitary tumor with marked contrast enhancement is seen in the sella turcica, compressing the optic nerve superiorly. (**b**) MRI obtained during the interval between the initial and second pituitary tumor resections (7 days after the initial resection). Compression of the optic nerve by the tumor has been relieved, although residual tumor tissue remains, predominantly on the left side. (**c**) MRI obtained 400 days after the second pituitary tumor resection. The tumor has been completely resected, with no evidence of recurrence during follow-up.

**Figure 2 reports-08-00144-f002:**
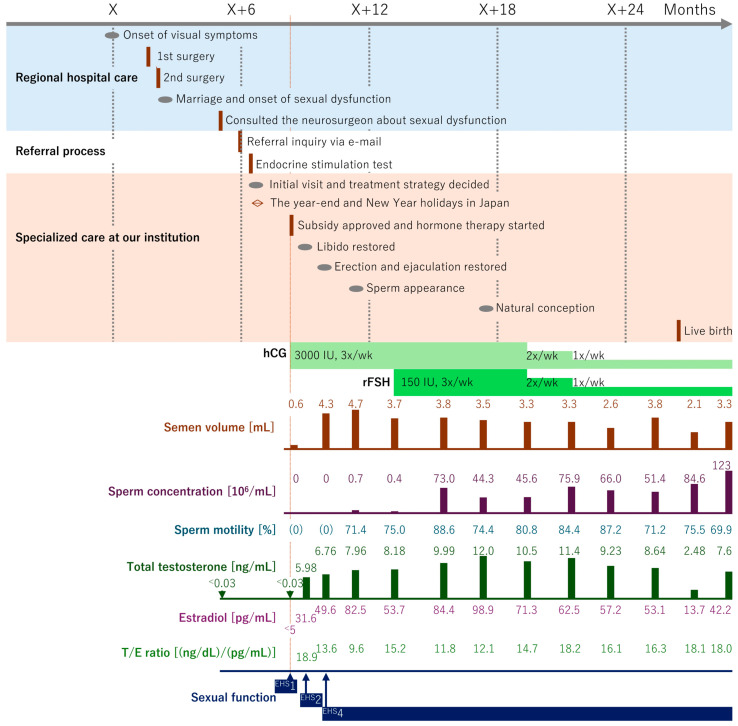
Timeline from the diagnosis of pituitary adenoma at a regional hospital through referral to our institution for specialized treatment, initiation of hormone therapy, and eventual live birth. The upper timeline illustrates the patient’s clinical course from initial diagnosis and referral through advanced treatment and achievement of pregnancy; Below the timeline, the dosages and timing of hormone replacement therapy—specifically human chorionic gonadotropin (hCG) and recombinant follicle-stimulating hormone (rFSH)—are shown, along with corresponding semen-analysis results, serum-sex-hormone concentrations, and changes in sexual function; Semen volume and sperm concentration are presented as both numerical values and bar graphs (reference values: ≥1.6 mL for semen volume, ≥16 × 106/mL for sperm concentration), while sperm motility is shown numerically only (reference value: ≥42%); Total testosterone levels are presented both numerically and as bar graphs (reference interval: 1.31–8.7 ng/mL), whereas estradiol concentrations are presented numerically only (reference interval: 14.6–48.8 pg/mL). Once both total testosterone and estradiol levels exceed the assay detection limits, the ratio of total testosterone (converted from ng/mL to ng/dL) to estradiol (pg/mL) is calculated and displayed as the T/E ratio; Sexual function is indicated by the Erection Hardness Score (EHS), graded on a four-point scale (1 = lowest, 4 = highest).

**Figure 3 reports-08-00144-f003:**
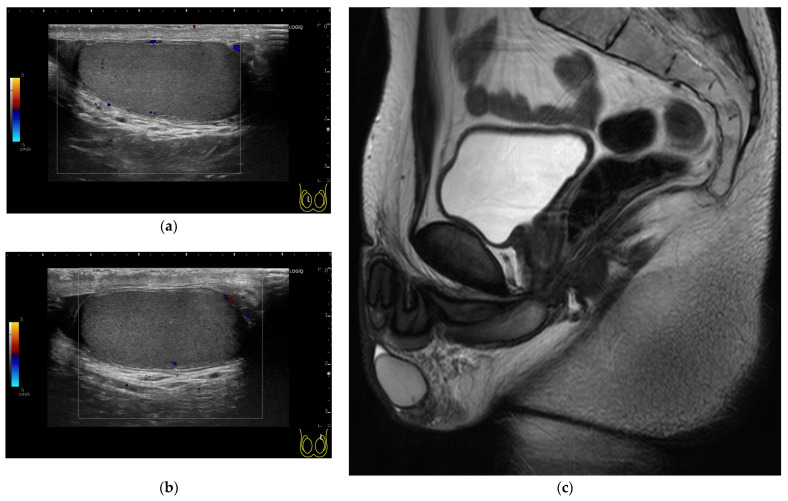
Images of the genitalia just before initiation of hormone replacement therapy after two pituitary resections. Color Doppler ultrasonography showing longitudinal sections of the testicular parenchyma ((**a**) right testis, (**b**) left testis). Sagittal T2-weighted image of pelvic organs on plain MRI (**c**).

**Table 1 reports-08-00144-t001:** Changes in pituitary, thyroid, adrenal, and gonadal hormone levels at various time points before and after the first and second surgeries.

	Pre-op	Post-1st op	Post-2nd op (Days from 1st op in Parentheses)			LLR	ULR
		POD 6	POD 19 (33)	POD 47 (61)	POD 89 (103)		
GH	0.47	0.90	0.26	0.19	0.16	0.03	2.47
IGF-1	113	138	64	47	60	111	309
PRL	38.1	19.2	19.2	17.8	19.5	4.29	13.69
TSH	0.57	0.01	0.06	0.07	0.03	0.50	5.00
FT3	3.28	2.55	2.54	3.20	4.17	2.30	4.00
FT4	1.39	1.11	0.69	0.74	0.91	0.90	1.70
ACTH	26.2	4.6	1.6	4.7	1.9	7.2	63.3
Cortisol	8.3	0.5	0.1	0.2	0.2	7.07	19.6
LH	1.58	0.12	<0.10	0.13	0.12	0.79	5.72
FSH	21.40	1.99	0.34	0.37	0.46	2.00	8.30
TT	NA	NA	NA	NA	<0.03	1.31	8.71

Pre-op, before surgery (baseline at 15 days before the 1st surgery); post-1st/2nd op, after the 1st/2nd surgery; POD postoperative day; LLR, lower limit of reference intervals; ULR, upper limit of reference intervals; GH, growth hormone [ng/mL]; IGF-1, insulin-like growth factor 1 [ng/mL]; PRL, prolactin [ng/mL]; TSH, thyroid-stimulating hormone [μIU/mL]; FT3, free triiodothyronine [pg/mL]; FT4, free thyroxine [ng/dL]; ACTH, adrenocorticotropic hormone [pg/mL]; cortisol [μg/dL]; LH, luteinizing hormone [mIU/mL]; FSH, follicle-stimulating hormone [mIU/mL]; TT, total testosterone [ng/mL]; NA, not available.

**Table 2 reports-08-00144-t002:** Representative Japanese guidelines related to HH.

Title	Main Publisher	Year	References
[Guidelines for the Diagnosis and Treatment of Gonadotropin Deficiency] ^†^	JES, JHPT	2010	[20]
		2019	[21]
		2023	[22]
[Clinical Practice Manual for Late-Onset Hypogonadism]	JUA, JSMH	2007	[23,24]
		2022	[25,26]
[Guidelines for Male Hypogonadism]	JES, JSMH	2022	[27]
[Clinical Practice Guidelines for Male Infertility]	JUA, JSRM	2024	[28,29]
[Clinical Practice Guidelines for Differences of Sex Development]	JSPE, JES, JSPU, JSRE, JSGI	2025	[30]

Titles enclosed in brackets represent translations of Japanese titles and may not necessarily reflect the official English names. ^†^ This guideline was compiled by investigators from the Survey and Research Group for Hypothalamic-Pituitary Dysfunction (Health and Labour Sciences Research Grant for Research on Rare and Intractable Diseases), with the cooperation of JES and JHPT. HH, hypogonadotropic hypogonadism; JES, the Japan Endocrine Society; JSHP, the Japanese Society for Hypothalamic and Pituitary Tumors; JUA, the Japanese Urological Association; JSMH, the Japanese Society of Men’s Health; JSRM, the Japan Society for Reproductive Medicine; JSPE, the Japanese Society for Pediatric Endocrinology; JSPU, the Japanese Society of Pediatric Urology; JSRE, the Japan Society of Reproductive Endocrinology; and JSGI, the Japanese Society of Gender Incongruence.

**Table 3 reports-08-00144-t003:** Sequential changes in anterior pituitary hormone concentrations following stimulation with GHRP-2, TRH, CRH, and GnRH after two pituitary resections.

	Pre	15 min	30 min	45 min	60 min	90 min	120 min	Peak/Pre
GH	0.24	0.20	0.23	0.33	0.40 ^†^	0.35	0.25	1.67
PRL	19.7	21.5	22.7 ^†^	22.2	22.0	20.5	19.6	1.15
TSH	0.26	0.47	0.84	1.08	1.13 ^†^	1.13 ^†^	1.01	4.35
ACTH	6.6	54.2	58.3 ^†^	55.0	37.6	25.8	22.2	8.83
LH	0.18	0.37	0.54	0.66	0.67 ^†^	0.62	0.54	3.72
FSH	0.55	0.66	0.78	0.87	1.00	1.02	1.04 ^‡^	1.89

Serum concentrations of anterior pituitary hormones before stimulation (pre), and at 15, 30, 45, 60, 90, and 120 min after stimulation are shown. ^†^ indicates the time point at which each hormone reached its peak concentration. ^‡^ indicates that the hormone continued to rise throughout the 120‑min observation period and no peak was observed. “Peak/pre” represents the ratio of the peak serum concentration to the pre-stimulation value. The growth hormone (GH) [ng/mL] row shows changes in serum GH in response to stimulation with regular insulin at a dose of 0.1 U/kg body weight [44]. The peak concentration after stimulation did not increase sufficiently, suggesting pituitary GH deficiency. According to Japanese guidelines, GH deficiency is defined as a peak ≤ 3 ng/mL at pre-stimulation, 30, 60, 90, or 120 min [21,22]; The prolactin (PRL) [ng/mL] and thyroid-stimulating hormone (TSH) [μIU/mL] rows show the responses to 500 μg protirelin (thyrotropin-releasing hormone, TRH). Pre-stimulation PRL was slightly above the upper limit of reference intervals (ULR, 13.69 ng/mL) but below the threshold of 200 ng/mL for prolactinoma [45]. This is consistent with previous reports of the postoperative course after the resection of non-functional pituitary adenoma [46]. The mildly elevated PRL may be due to impaired hypothalamic secretion of prolactin-inhibiting factor (PIF) caused by the tumor, with persistent effects after resection. PRL showed almost no response to stimulation (a peak > 30 ng/mL or peak/pre ratio > 2 is considered a positive response [47]). TSH was below the lower limit of the reference interval (LLR, 0.50 μIU/mL) at baseline, and the post-stimulation peak did not sufficiently increase, suggesting pituitary TSH deficiency (a peak ≥ 5.5 μIU/mL [48] or >6.0 μIU/mL [47] is considered a positive response). Although the TSH peak normally occurs at 30–60 min after stimulation [49], in this case, the peak was delayed until 90–120 min. This may reflect not only pituitary dysfunction but also impaired physiological TRH secretion from the hypothalamus [50]; The adrenocorticotropic hormone (ACTH) [pg/mL] row shows the response to 100 μg corticorelin (corticotropin-releasing hormone, CRH). Pre-stimulation ACTH was below the LLR (7.2 pg/mL), but the response was sufficient. A peak between 30 and 60 min and a peak/pre ratio ≥ 2 is considered a positive response [43,51]. This suggests hypothalamic adrenal insufficiency [52]; The luteinizing hormone (LH) [mIU/mL] and follicle-stimulating hormone (FSH) [mIU/mL] rows show responses to 100 μg gonadorelin (gonadotropin-releasing hormone, GnRH). Pre-stimulation LH and FSH were below their respective LLRs (0.79 mIU/mL and 2.00 mIU/mL). LH showed an insufficient response, suggesting pituitary LH deficiency. FSH was also extremely low at baseline, indicating pituitary FSH deficiency, though some response to stimulation was observed. However, FSH did not peak within 120 min, indicating a delayed response. A peak/pre ratio > 5 for LH and >1.5 for FSH is considered a positive response [47]. Normally, LH and FSH peaks are observed at 30 and 60 min, respectively [53]. Such delays may result from both pituitary dysfunction and impaired physiological GnRH secretion from the hypothalamus [54].

**Table 4 reports-08-00144-t004:** Sequential changes in serum-sex-hormone concentrations following stimulation with hCG after two pituitary resections.

	Day 1 (Pre-hCG)	Day 2	Day 3	Day 4	LLR	ULR
TT	<0.03	0.83	2.99	3.87	1.31	8.71
FT	<0.2	1.6	5.1	8.5	7.6	23.8
E2	<5.0	6.6	14.5	14.4	14.6	48.8
T/E ratio	NA	12.6	20.6	26.9	NA	NA

This table shows the changes in serum concentrations of sex hormones in response to administration of human chorionic gonadotropin (hCG, 5000 U), a luteinizing hormone (LH) receptor agonist, given for three consecutive days. Measurements were taken on day 1 (pre-hCG, immediately before the first injection), day 2, day 3, and day 4 (the day after the final injection). Total testosterone (TT) [ng/mL]: The pre-hCG value was below the detection limit (<0.03 ng/mL), but increased to approximately 28 times the detection limit by day 2, and continued to rise, reaching about three times the lower limit of the reference interval (LLR; 1.31 ng/mL) by day 4. A doubling or greater increase in TT over the 4 days of the hCG stimulation test is considered a minimum criterion for response [17,58]. In prepubertal children, a post-hCG TT level of ≥1.1 ng/mL has been proposed as a cutoff for the indication of hormone replacement therapy [62]; however, no established consensus cutoff for adult-onset hypogonadotropic hypogonadism (HH) was observed; Free testosterone (FT) [pg/mL] was measured by the radioimmunoassay method [63]. Although not routinely measured during the hCG stimulation test, FT is commonly used as an indicator (cutoff: 7.5 pg/mL) in the diagnosis of late-onset hypogonadism in Japan and helps interpret clinical symptoms of testosterone deficiency in adults [26]. It is therefore shown here for reference; Estradiol (E2) [pg/mL]: E2 is also not routinely measured during the hCG stimulation test. In this table, the ratio of total testosterone (converted from ng/mL to ng/dL) to estradiol (pg/mL), i.e., the T/E ratio, is shown for reference, since an association between this ratio and sexual function has been suggested (cutoff: 12.0) [64]. NA, not available.

**Table 5 reports-08-00144-t005:** Sequential changes in AMS and EHS, and baseline IIEF-5 and ejaculatory function, before and after HRT.

Items	No.	Symptoms	Pre	Day 42	Day 84
AMS-som	1	Impaired well-being	1	NA	1
AMS-som	2	Joint complaints	1	NA	1
AMS-som	3	Excessive sweating	1	NA	1
AMS-som	4	Frequent sleep disturbance	1	NA	1
AMS-som	5	Need for sleep	1	NA	1
AMS-psy	6	Irritability	2	NA	1
AMS-psy	7	Nervousness	2	NA	1
AMS-psy	8	Anxiety	1	NA	1
AMS-som	9	Physical exhaustion	1	NA	1
AMS-som	10	Muscular weakness	2	NA	1
AMS-psy	11	Depressive mood	1	NA	1
AMS-sex	12	Having passed the peak	1	NA	1
AMS-psy	13	Feeling burnt out	1	NA	1
AMS-sex	14	Decreased beard growth	2	NA	1
AMS-sex	15	Impaired sexual potency	2	NA	1
AMS-sex	16	Less frequent morning erections	4	NA	1
AMS-sex	17	Disturbed libido	4	NA	1
AMS		Somatic subscale (7–35)	8	NA	7
AMS		Psychological subscale (5–25)	7	NA	5
AMS		Sexual subscale (5–25)	13	NA	5
AMS		Total (17–85)	28	NA	17
EHS			1	2	4
IIEF-5	1	Erection confidence	1	NA	NA
IIEF-5	2	Erection hardness	1	NA	NA
IIEF-5	3	Erection maintenance	1	NA	NA
IIEF-5	4	Maintenance difficulty	1	NA	NA
IIEF-5	5	Intercourse satisfaction	1	NA	NA
IIEF-5		Total (5–25)	5	NA	NA
MSHQ-EjD-SF	1	Ejaculation frequency	1	NA	NA
Semen volume [mL]			0.6 ^†^	NA	4.3

This table shows the changes in the patient’s sexual function before and after hormone replacement therapy (HRT), based on responses to questionnaires routinely used at our institution. NA, not available. The patient completed the Japanese version of the Heinemann Aging Male’s Symptoms (AMS) Scale [70] before HRT (pre) and at 84 days after HRT initiation. The AMS is a 17-item instrument rated on a five-point Likert scale from 1 (no symptom) to 5 (very severe symptoms), with higher cumulative scores indicating greater severity. The total AMS score ranges from 17 to 85 and is classified as follows: no/little complaints (17–26), mild (27–36), moderate (37–49), and severe complaints (50–85) [71]; Each AMS item can be categorized into the somatic (seven items), psychological (five items), or sexual (five items) subscales [72]. Although no universal classification exists for the severity of each subscale, previous studies have proposed the following groupings: somatic subscale—none or mild (7–12), moderate (13–18), severe (19–35); psychological subscale—none or mild (5–8), moderate (9–12), severe (13–25); and sexual subscale—none or mild (5–7), moderate (8–10), severe (9–25) [73]; The patient also completed the Erection Hardness Score (EHS) [74] before HRT (pre), at 42 days and at 84 days after HRT initiation. This self-reported tool, widely used in clinical practice, evaluates erection hardness on a four-point Likert scale, with grade 1 indicating the least hard and grade 4 indicating the hardest [75]; The patient completed the International Index of Erectile Function-5 (IIEF-5) [76] before HRT initiation. The IIEF-5 consists of five items rated on a five-point Likert scale; lower cumulative scores indicate more severe erectile dysfunction. Severity is classified as follows: severe (5–7), moderate (8–11), mild to moderate (12–16), mild (17–21), and no erectile dysfunction (22–25). The IIEF-5 assesses symptoms over the previous 6 months; since erectile function was fully restored by 84 days after HRT initiation, and to avoid confusion for the patient, the IIEF-5 was not administered after HRT; The item in our questionnaire, “In the past month, how often have you been able to ejaculate or ‘cum’ when having sexual activity?” was answered by the patient before HRT initiation. This item is rated on a five-point Likert scale from 1 (could not ejaculate) to 5 (all the time), and is identical to the first question of the Male Sexual Health Questionnaire-Ejaculatory Dysfunction-Short Form (MSHQ-EjD-SF) [77]. ^†^ Before HRT, despite prolonged masturbation, the fluid collected in a special container for semen analysis consisted only of mucus secreted without any sensation of ejaculation or orgasm.

## Data Availability

The data are contained within the article.

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
