# Peer review of "Comprehensive Fertility Management After Pituitary Adenoma Surgery: Lessons from a Rural Japanese Case and Practical Review"

_reports, 2025, doi:10.3390/reports8030144_

Round 1

Reviewer 1 Report

Comments and Suggestions for Authors

First, I would like to thank for giving the opportunity of reviewing this paper entitled “Comprehensive Fertility Management After Pituitary Adenoma Surgery: Lessons from a Rural Japanese Case and Practical Review” from Japan. This article sections are organized and the study design is straightforward. The objective of this study report have been clearly explained.

The case report have been written in detail, but the authors ignored diagnostic limitations. The authors should mention the diagnostic limitations of this case.

Antimullerian hormone and inhibin B provide significant knowledge regarding increasing Sertoli cell mass and function. AMH and inhibin B hormone have great importance in the pre- and post-recombinant hormone therapy follw-up, because these hormones are used for monitoring the hypogonadotropic hypogonadism therapy. The authors did not mention the importance of these markers in monitoring treatment in detail.

Is the serum estradiol level for this patient within the normal reference values for a healthy person after hormone replacement therapy? The authors should also discuss the subject between high serum serum estradiol level and cancer and cardiovascular disease.

In the case presentation, it should be explained why a gamma knife radiosurgery was not preferred for the treatment of pituitary adenoma. Gamma knife radiosurgery is a noninvasive therapy for pituitary adenoma that allows physicians to preserve most of the patient’s healthy pituitary tissue. This game-changing procedure has been especially effective for pituitary adenomas. The authors should consider this therapy before first transnasal-sphenoidal surgery approach.

English language fine. No issues detected.

Author Response

Comment 1:

First, I would like to thank for giving the opportunity of reviewing this paper entitled “Comprehensive Fertility Management After Pituitary Adenoma Surgery: Lessons from a Rural Japanese Case and Practical Review” from Japan. This article sections are organized and the study design is straightforward. The objective of this study report have been clearly explained.

Response 1:

We would like to sincerely thank you for taking the time out of your busy schedule to review our manuscript. We are truly grateful for your careful reading and for your generous overall evaluation of our work. We will now address each of your comments in detail, one by one, below.

Comment 2:

The case report have been written in detail, but the authors ignored diagnostic limitations. The authors should mention the diagnostic limitations of this case.

Response 2:

Thank you for your valuable comment. In alignment with the feedback from other reviewers, we have added a description of the strengths and limitations of this case report to the end of the Discussion section (Lines 591–609).

Comment 3:

Antimullerian hormone and inhibin B provide significant knowledge regarding increasing Sertoli cell mass and function. AMH and inhibin B hormone have great importance in the pre- and post-recombinant hormone therapy follw-up, because these hormones are used for monitoring the hypogonadotropic hypogonadism therapy. The authors did not mention the importance of these markers in monitoring treatment in detail.

Response 3:

Thank you for highlighting this important point. We fully agree that AMH and inhibin B provide highly valuable information for monitoring therapy in hypogonadotropic hypogonadism. Unfortunately, in Japan, the measurement of AMH and inhibin B is not routinely performed in actual clinical practice, and these markers were not available for monitoring in the present case. In the manuscript, we have addressed the importance of these markers in the paragraph immediately preceding Table 4, along with a brief explanation of the circumstances in Japan (Lines 308–316).

Comment 4:

Is the serum estradiol level for this patient within the normal reference values for a healthy person after hormone replacement therapy? The authors should also discuss the subject between high serum serum estradiol level and cancer and cardiovascular disease.

Response 4:

Thank you for your insightful comment. As you have pointed out, the patient’s serum estradiol level exceeded the normal reference values. This is noted in the manuscript as follows: “E2 levels, which often exceeded the upper limit of the reference interval (48.8 pg/mL) due to hCG stimulation” (Line 473). In addition, we have added a brief discussion regarding cancer and cardiovascular disease to the penultimate paragraph of the Case Presentation section (Line 475).

Comment 5:

In the case presentation, it should be explained why a gamma knife radiosurgery was not preferred for the treatment of pituitary adenoma. Gamma knife radiosurgery is a noninvasive therapy for pituitary adenoma that allows physicians to preserve most of the patient’s healthy pituitary tissue. This game-changing procedure has been especially effective for pituitary adenomas. The authors should consider this therapy before first transnasal-sphenoidal surgery approach.

Response 5:

Thank you for your valuable comment. As you have noted, gamma knife radiosurgery can be considered a potential alternative to surgical intervention. In this case, however, gamma knife radiosurgery was not selected, in accordance with the general consensus in Japan that surgical resection remains the first-line option as long as the tumor is resectable. We have added this consensus, along with a discussion of the usefulness of gamma knife radiosurgery and relevant citations, in the paragraph following Table 1 (Lines 134–139).

Comment 6:

English language fine. No issues detected.

Response 6:

Thank you very much. We appreciate your kind feedback and look forward to your continued guidance.

Reviewer 2 Report

Comments and Suggestions for Authors

This report does not add novelty and is a simple description of an ordinary case of HH post-surgery. It lacks of practical aspects and is not useful to the current literature on this topic

Author Response

Comment 1:

This report does not add novelty and is a simple description of an ordinary case of HH post-surgery.

Response 1:

We sincerely thank you for taking the time to review our manuscript. We also wish to express our apologies for any differences in perception that may have arisen due to insufficient explanation on our part. As you have pointed out, the pathophysiology, diagnostic methods, and treatment approach described in this case are all routinely encountered in clinical practice.

The novelty of our case report lies, first, in being the first documented case in which the reasons why the patient’s personal priorities were entirely overlooked during tumor treatment were explicitly described. We believe this provides valuable insight into previously reported gaps in awareness of fertility preservation guidelines among tumor specialists, even after such guidelines have been published.

Second, we consider it of particular value that this case provides real-world evidence suggesting the importance of considering fertility preservation and providing appropriate counseling even for benign tumors that do not require chemotherapy, before initiating any tumor-directed intervention.

To address the importance of these points more clearly, we have added descriptions to the Abstract (Lines 35–39), at the end of the Introduction section (Lines 73–78), and at the end of the Discussion section (Lines 591–600). We would be most grateful if you could kindly review the revised manuscript again in light of these clarifications.

Comment 2:

It lacks of practical aspects and is not useful to the current literature on this topic.

Response 2:

We sincerely appreciate your candid feedback and take your critical comment to heart. We endeavored to conduct a literature review as aligned as possible with practical aspects, supplementing it with our own perspectives while presenting the case. In the current revision, we have further strengthened areas previously identified by other reviewers as insufficiently explained.

That said, as this manuscript is primarily a case description, we recognize that there are inherent limitations in overgeneralizing its findings. We have explicitly acknowledged this point in the final part of the Discussion section (Lines 601–609).

Furthermore, we are currently planning a broader narrative review focusing on fertility preservation and the development of oncofertility networks, not limited to pituitary adenomas but encompassing benign diseases more generally. We are committed to publishing such work in the future to complement the present manuscript.

Reviewer 3 Report

Comments and Suggestions for Authors

The article Comprehensive Fertility Management After Pituitary Adenoma Surgery: Lessons from a Rural Japanese Case and Practical Review presents an interesting case of a patient with PA, who underwent two surgeries, complicated by HH. Nevertheless, medical procedures allowed him to regain proper fertility. Moreover, authors present short review of the literature regarding the topic.

Abstract:

Has to be more coherent, please remove all unnecessary comments (like one about institutional holidays and geographic barriers).

Introduction:

line 66 – what is “Note 1”

Case presentation:

The case description is extensive and thorough, with lots of details – which is of course good – however, attempts could be made to reduce its volume to improve readability.

Discussion:

Is well-written and concise case study.

Conclusions:

They summarize the article well.

Overall the article is very well and thoroughly prepared, I don't detect any major flaws.

With only minor revisions it could be suitable for publication.

Author Response

Comment 1:

The article Comprehensive Fertility Management After Pituitary Adenoma Surgery: Lessons from a Rural Japanese Case and Practical Review presents an interesting case of a patient with PA, who underwent two surgeries, complicated by HH. Nevertheless, medical procedures allowed him to regain proper fertility. Moreover, authors present short review of the literature regarding the topic.

Response 1:

We are deeply grateful for the valuable opportunity to have our manuscript reviewed. We will address each of your comments in detail below.

Comment 2:

Abstract:

Has to be more coherent, please remove all unnecessary comments (like one about institutional holidays and geographic barriers).

Response 2:

Thank you for your helpful suggestion. In accordance with your advice, we have identified and removed unnecessary details from the Abstract to enhance its coherence, and we have revised the wording to make the focus of the study clearer.

Comment 3:

Introduction:

line 66 – what is “Note 1”

Response 3:

Thank you for pointing this out. This was an extraneous text element that appeared as a result of multiple revisions. We will take care to avoid such oversights in the future.

Comment 4:

Case presentation:

The case description is extensive and thorough, with lots of details – which is of course good – however, attempts could be made to reduce its volume to improve readability.

Response 4:

Thank you for your candid comment. I fully agree with your observation that the manuscript is overly detailed and lengthy, and I would first like to apologize for not being able to reduce its volume on this occasion, despite recognizing the importance of improving readability. The hesitation to remove certain content stems from the circumstances of its writing and revision process.

This manuscript was originally submitted to a different journal, and following an open peer review process, the publisher, MDPI, recommended resubmission to the current journal. During the initial submission, we received feedback from four reviewers, addressing various aspects of the work, which led us to make substantial additions to the text. While the original manuscript was considerably shorter, the revisions ultimately expanded it to its current length. However, we believe that each portion of the present content retains potential value for readers of this journal. We hope for your kind understanding in this regard.

For reference, we have shared the content of the previous open peer review process for your consideration.

Comment 5:

Discussion:

Is well-written and concise case study.

Response 5:

Thank you for your kind feedback. In line with comments from other reviewers, we have made minor additions to this section.

Comment 6:

Conclusions:

They summarize the article well.

Response 6:

We sincerely appreciate your positive evaluation.

Comment 7:

Overall the article is very well and thoroughly prepared, I don't detect any major flaws.

With only minor revisions it could be suitable for publication.

Response 7:

Once again, we are grateful for the opportunity to have our manuscript reviewed. We look forward to your continued guidance.

Round 2

Reviewer 1 Report

Comments and Suggestions for Authors

None

Reviewer 2 Report

Comments and Suggestions for Authors

Modest improvement of the qulity of the manuscript after R1

Reviewer 3 Report

Comments and Suggestions for Authors

Authors has revised the manuscript, and answered all queries. I do not have any further comments.